# PARETO-GUIDED REGIONAL SAMPLING FOR ADAPTIVE COLLOCATION IN PHYSICS-INFORMED NEURAL NETWORKS

## ABSTRACT

Physics-informed neural networks (PINNs) provide a powerful framework for solving both forward and inverse problems of differential equations by embedding physical constraints in the loss function. However, in regions that exhibit localized stiffness or sharp gradients, conventional PINN training using uniformly sampled collocation points often leads to suboptimal training efficiency and predictive accuracy. To address this challenge, this work proposes a Pareto-guided regional sampling (PaRS) framework for adaptive collocation point sampling and tuning. The proposed method integrates three key components: residual decomposition, adaptive loss weighting, and dynamic resampling. Specifically, the spatial domain of collocation points is first partitioned into multiple subregions, and the training objectives, such as residuals from different subregions, are treated as competing objectives. As a result, a Pareto front is constructed to capture trade-offs among competing residual and other losses. Subsequently, a novel Pareto-guided weighting method is developed to assign adaptive weights to each objective, informed by training progress and information across the Pareto front. These weights further guide the resampling of collocation points in each region, which increases the sampling density in underexplored or error-prone regions. The results of the experiment demonstrate that the proposed PaRS-PINN performs better than standard PINNs and state-of-the-art adaptive collocation point methods.

## 1 INTRODUCTION

Physics-informed neural networks (PINNs) have recently been powerful tools to solve both forward and inverse problems governed by differential equations (Hao et al., 2024). PINNs bridge the gap between data-driven models and physics-based models by incorporating governing equations directly into the loss of neural networks. Compared to conventional machine learning models that may require extensive labeled datasets and often overlook physical feasibility, PINNs are well-suited for problems characterized by scarce data, complex boundary conditions, or noisy observations. This distinctive advantage has fueled growing interest in diverse domains, including heat transfer (Cai et al., 2021b), medical application (Rodríguez et al., 2023), fluid dynamics (Chen et al., 2025; Cai et al., 2021a), and various engineering applications (Karniadakis et al., 2021; Zobeiry & Humfeld, 2021).

Although PINNs have achieved considerable success, they still face challenges in solving complex partial differential equations (PDEs), as the optimization process may become trapped in local minima (Krishnapriyan et al., 2021; Wang et al., 2021; 2022). A critical factor influencing the effectiveness of PINN training is the selection and distribution of collocation points, which serve as enforcement locations for the governing equations (Wu et al., 2023; Matsubara & Yaguchi, 2025). Conventional approaches typically adopt static and uniformly distributed collocation points throughout the spatiotemporal domain. However, in many real-world systems, the solution may exhibit spatial and temporal heterogeneity, such as steep gradients or localized stiffness. When all residual points are treated equally, the PINN model may allocate excessive modeling capacity to smooth regions while failing to adequately capture regions with high residual errors or dynamic complexities, which can hinder both training efficiency and solution accuracy.

Recent research has concentrated on the development of adaptive sampling strategies for collocation points in PINNs (Hanna et al., 2022; Nabian et al., 2021; Matsubara & Yaguchi, 2025; Yang et al., 2023). For example, a residual-based adaptive refinement (RAR) strategy is proposed to add new collocation points in locations showing large residuals (Lu et al., 2021). A comprehensive study is conducted to compare non-adaptive and residual-based adaptive sampling strategies in PINNs, which also introduced residual-based adaptive distribution (RAD) and residual-based adaptive refinement with distribution (RAR-D) methods. Additionally, a novel Retain-Resample-Release sampling (R3) algorithm is proposed to incrementally retain collocation points in regions of high residuals and uniformly resample new points (Daw et al., 2023). However, most existing adaptive sampling methods treat collocation points independently without considering the spatial structure and complex interactions between different collocation points. These methods tend to sample regions with high residuals while neglecting other areas that may also be critical for model generalization, limiting their ability to systematically balance learning across entire domains.

Another significant factor that affects the efficiency of the PINN training is the allocation of weights in different loss terms (Rathore et al., 2024; Hwang & Lim, 2024). Although several studies have investigated strategies for modifying the relative contributions among different loss terms (i.e., residual loss, initial condition loss, boundary condition loss, and data loss) (McClenny & Braga-Neto, 2023; Liu et al., 2024; Xiang et al., 2022; Bischof & Kraus, 2025), little attention has been paid to the internal imbalance within the residual loss itself, especially when collocation points are unevenly distributed across regions with different dynamics. The learning process of PINNs can be naturally framed as a multi-objective optimization (MOO) task from an optimization standpoint, in which each individual loss component corresponds to a distinct objective function to be optimized. This view allows MOO techniques to explore the solution landscape more comprehensively and derive Pareto front insights that highlight trade-offs among objectives (Gunantara, 2018). Despite this potential, there has been limited research on adaptively adapting the sampling and weights of collocation points so far. Addressing this gap offers an important opportunity to improve both the efficiency and accuracy of PINN training by systematically refining the treatment of collocation points.

Motivated by the above considerations, this work proposes a novel Pareto-guided regional sampling (PaRS) framework for jointly sampling and tuning the weights of collocation points. Specifically, we first decompose the residual loss into multiple sub-objectives by partitioning the spatial domain into distinct regions. An MOO algorithm is implemented to explore the Pareto front that reflects different trade-offs among regional residuals, initial/boundary losses, and data loss. To adaptively guide training, we develop a Pareto-guided weighting scheme that dynamically assigns loss weights based on the information provided by Pareto front. These weights are further used to inform a resampling strategy, where the collocation point density in each region is adjusted to reflect its current learning importance. The capability of the proposed PaRS framework to improve the performance of PINNs is validated through a series of case studies including both partial and ordinary differential equations.

## 2 Physics-informed Neural Network

Consider a general nonlinear partial differential equation (PDE) described as:

$$
\begin{aligned}
\mathcal{A}(u(x,t);k) &= f(x,t), & x \in \Phi, t \in [0,T], \\
\mathcal{B}(u(x,t)) &= g(x,t), & x \in \partial\Phi, t \in [0,T], \\
\mathcal{C}(u(x,0)) &= h(x), & x \in \Phi,
\end{aligned}
\tag{1}
$$

where $t$ and $x$ denote the time and spatial coordinates, respectively. $\Phi$ is the spatial domain, $\partial\Phi$ is the boundary of spatial domain, $T$ is the time horizon. $u(x,t)$ is the latent solution to be discovered, constrained by the boundary condition $\mathcal{B}$ and initial condition $\mathcal{C}$. $\mathcal{A}$ is a differential operator. $k$ denotes a set of unknown parameters, and $f$,$g$,$h$ are three known functions. The approximation of the solution is learned by a neural network $\hat{u}(x,t;\theta,k)$ in PINNs, where $\theta$ and $k$ denote the trainable parameters. The expressions of the loss terms can be represented as:

$$
\begin{aligned}
\mathcal{L}_r(\theta,k) &= \frac{1}{N_r} \sum_{j=1}^{N_r} \left\| \mathcal{A}(\hat{u}(x_r^j, t_r^j; \theta, k)) - f(x_r^j, t_r^j)] \right\|^2, \\
\mathcal{L}_{BC}(\theta,k) &= \frac{1}{N_b} \sum_{j=1}^{N_b} \left\| \mathcal{B}(\hat{u}(x_b^j, t_b^j; \theta, k)) - g(x_b^j, t_b^j) \right\|_2^2, \\
\mathcal{L}_{IC}(\theta,k) &= \frac{1}{N_i} \sum_{j=1}^{N_i} \left\| \mathcal{C}(\hat{u}(x_i^j, 0; \theta, k)) - h(x_i^j) \right\|_2^2, \\
\mathcal{L}_{data}(\theta,k) &= \frac{1}{N_{data}} \sum_{i=1}^{N_{data}} \left\| \hat{u}\left(x_{data}^j, t_{data}^j; \theta, k\right) - u_{data}^j \right\|_2^2,
\end{aligned}
\tag{2}
$$

where $\mathcal{L}_r$ is the residual loss term, which quantifies how well the predicted solution matches the governing PDEs at interior collocation points $\{(x_r^j, t_r^j)\}_{j=1}^{N_r}$. The initial condition loss $\mathcal{L}_{IC}$ and the boundary condition loss $\mathcal{L}_{BC}$ render the solution to the specified initial data $\{x_i^j, u(x_i^j, 0)\}_{j=1}^{N_i}$ and boundary data $\{(x_b^j, t_b^j), u(x_b^j, t_b^j)\}_{j=1}^{N_b}$, respectively. Furthermore, when additional measurements are available, the data loss $\mathcal{L}_{data}$ can be incorporated to assess the difference between the predicted solution and observed data $\{x_{data}^j, t_{data}^j, u_{data}^j\}_{j=1}^{N_{data}}$.

## 3 PARETO-GUIDED REGIONAL SAMPLING (PARS)

In standard PINN frameworks, the residual loss $\mathcal{L}_r$ is typically computed over a global set of collocation points uniformly sampled in the spatiotemporal domain $\Omega \times [0, T]$. However, such uniform treatment may obscure local discrepancies and prevent the model from focusing on regions with larger approximation errors or more complex dynamics. Therefore, this section introduces a Pareto-guided regional sampling (PaRS) framework to decompose the residual loss into regional components and systematically determine the loss weights and sampling strategy associated with different regions during training. The overall framework of PaRS is illustrated in Fig. 1.

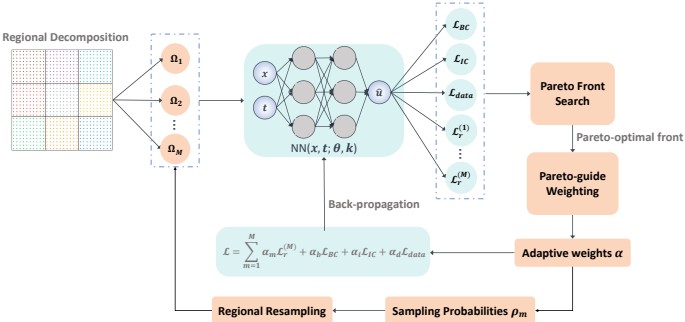

Figure 1: Schematic of Pareto-guided regional sampling (PaRS) framework for PINN training.

### 3.1 REGIONAL DECOMPOSITION

We first decompose the global collocation point set into multiple non-overlapping subdomains and then assign a shared weight to each region instead of individual points, which allows the model to explicitly account for spatial heterogeneity during training and maintain a manageable number of trainable loss weights. Specifically, the entire set of residual points $\mathcal{X}_r = \{(x_r^j, t_r^j)\}_{j=1}^{N_r}$ is divided into $M$ sub-regions $\{\Omega_m\}_{m=1}^M$, such that $\bigcup_{m=1}^M \Omega_m = \mathcal{X}_r$ and $\Omega_m \cap \Omega_{m'} = \emptyset$ for $m \neq m'$. Each region $\Omega_m$ contains a subset of collocation points $\{(x_r^{j,m}, t_r^{j,m})\}_{j=1}^{N_m}$, where $N_m$ is the number of collocation points assigned to subdomain $m$, satisfying $\sum_{m=1}^M N_m = N_r$. While it is theoretically possible to assign a distinct weight to each individual collocation point, it would introduce an excessive number of weighting parameters and result in high computational cost or unstable optimization.

In this work, we employ K-Means clustering (Likas et al., 2003) on normalized collocation coordinates to identify these subregions. This clustering algorithm groups collocation points into clusters based on spatial-temporal proximity, resulting in subdomains that reflect the geometry and distribution of the original collocation point set. Compared to the uniform grids method and Latin Hypercube Sampling (LHS), K-means is more suitable for high-dimensional domains and can generate deterministic and compact partitions that naturally align with spatial geometry. Once the decomposition is complete, the residual loss for each subdomain is defined as follows:

$$\mathcal{L}_r^{(m)}(\theta, k) = \frac{1}{N_m} \sum_{j=1}^{N_m} \left\| \frac{\partial \hat{u}}{\partial t}(x_r^{j,m}, t_r^{j,m}; \theta, k) + \mathcal{N}_x[\hat{u}(x_r^{j,m}, t_r^{j,m}; \theta, k)] \right\|^2 \tag{3}$$

---

**Algorithm 1:** Pareto Front Search for Region-Wise Balancing

---

**Input:** $N_p$: population size; $N_g$: number of generations; $(\theta, k)$: current model parameters of
 PINN; $[\theta_{\min}, \theta_{\max}], [k_{\min}, k_{\max}]$: parameter bounds

**Step 1: Population Initialization**

Set $N_p$ candidate solutions $\{(\theta^i, k^i)\}_{i=1}^{N_p}$ as the current model parameters of PINN.

**Step 2: Evolutionary Loop**

**for** $g = 0$ **to** $N_g$ **do**

    **Evaluation Phase:**

    Evaluate each individual $(\theta^i, k^i)$ using the PINN model to compute the multi-objective

     vector: $\mathcal{F}^i = \left[ \mathcal{L}_r^{(1:M)}(\theta^i, k^i),\ \mathcal{L}_{IC}(\theta^i, k^i),\ \mathcal{L}_{BC}(\theta^i, k^i),\ \mathcal{L}_{\text{data}}(\theta^i, k^i) \right]$

    **Selection Phase:**

        Rank population $P_g$ based on non-dominated sorting

        Calculate crowding distance within each rank of $P_g$ to preserve diversity

        Select parent solutions based on a crowded-comparison operator

    **Variation Phase:**

        Apply crossover and mutation to generate an offspring population $Q_g$

        Merge $P_g$ and $Q_g$ to form the next population $P_{g+1}$

**Output:** Final Pareto front $\mathcal{F}$ from the evolved population $P_{N_g}$

---

The overall loss function is then formulated as a weighted sum of the individual components:

$$\mathcal{L}(\theta, k) = \sum_{m=1}^{M} \alpha_m \mathcal{L}_r^{(m)}(\theta, k) + \alpha_b \mathcal{L}_{BC}(\theta, k) + \alpha_i \mathcal{L}_{IC}(\theta, k) + \alpha_d \mathcal{L}_{data}(\theta, k) \quad (4)$$

where the weights $\alpha_m$ determine the relative contributions of each regional loss in the total physics loss. $\alpha_b$, $\alpha_i$, and $\alpha_d$ are loss weights used to balance the contributions of the initial, boundary, and data losses. Choosing appropriate values for these weights (i.e., $\alpha_m$, $\alpha_b$, $\alpha_i$, and $\alpha_d$) is critical, as poor balancing can hinder convergence or lead to suboptimal solutions.

## 3.2 PARETO FRONT SEARCH

Since the objectives in Eq. 4 are optimized jointly, the training procedure can be interpreted as an MOO task, aiming to identify solutions that achieve a trade-off among competing goals. The MOO problem can be formally expressed as:

$$\min_{\theta \in \Theta,\ k \in \mathcal{K}} \mathcal{L}(\theta, k) = \left[ \mathcal{L}_r^{(1)}(\theta, k),\ \mathcal{L}_r^{(2)}(\theta, k),\ \ldots,\ \mathcal{L}_r^{(M)}(\theta, k),\ \mathcal{L}_{IC}(\theta, k),\ \mathcal{L}_{BC}(\theta, k),\ \mathcal{L}_{\text{data}}(\theta, k) \right]^\top$$
$$(5)$$

where $\Theta$ and $\mathcal{K}$ are the feasible regions of the optimization problem. The solution of the MOO problem not only reflects the relative importance of each residual subregion but also coordinates the overall contribution of physics-informed terms and other loss terms. By decoupling these loss terms, the MOO framework allows the training process to adaptively explore trade-offs among conflicting objectives. Rather than searching for a single scalarized optimum, we aim to approximate the Pareto front of optimal compromises, from which adaptive weightings can be derived to guide the subsequent parameter updates of the PINN model.

Once the MOO formulation is established, the NSGA-II algorithm is implemented to explore a set of candidate solutions along the Pareto front. The NSGA-II algorithm (Deb et al., 2002) uses non-dominated sorting and crowding distance to efficiently approximate the Pareto front while preserving solution diversity throughout the optimization process. The steps for incorporating NSGA-II into the PINN training process are detailed in Algorithm 1. Specifically, the proposed algorithm initiates by constructing an initial population $P_0$, where each individual encodes a distinct pair of neural network parameters $\theta$ and unknown parameters $k$ used in the PINN model. Unlike the original NSGA-II procedure, which fully randomizes the initial population, the population $P_0$ in this work is initialized with the current parameter values from the ongoing PINN training.

During each generation $g$, the population undergoes an evaluation process. For each individual, its associated parameters $(\theta^i, k^i)$ are extracted and assigned to the PINN model. The model then evalu-

---

**Algorithm 2:** Pareto-Guided Weighting (PGW) Method

---

**Input:** Pareto front matrix $\mathcal{F} \in \mathbb{R}^{N_P \times N_o}$; historical losses $\{\mathcal{L}_{t,j}^{\text{hist}}\}$; window size $w$; data-term
       threshold $\epsilon$; temperature vector $\mathcal{T}_j$

**1. Normalization:** $F_{ij}^{norm} \leftarrow F_{ij}$

**2. Variability measure:** $\sigma_j \leftarrow F_{ij}^{norm}, \overline{F}_j^{norm}$

**3. Central tendency of current front:** $\overline{F}_j \leftarrow \frac{1}{N_p} \sum_{i=1}^{N_p} F_{ij}$

**4. Baseline extraction:** $\mathcal{L}_j^{\text{pre}} \leftarrow \{\mathcal{L}_{t,j}^{\text{hist}}\}, w$

**5. Relative improvement:** $r_j \leftarrow \overline{F}_j, \mathcal{L}_j^{pre}$

**6. Saliency score:** $\sigma_j \leftarrow F_{ij}^{norm}, \overline{F}_j^{norm}$

**7. Temperature adjustment:** $\mathcal{T}_{\text{data}} \leftarrow \epsilon$

**8. Weight computation:** $\alpha_j \leftarrow s_j, \mathcal{T}_j$

**Output:** Adaptive weights vector $\alpha \in \mathbb{R}^{N_o}$

---

ates the multiple objective terms, which collectively form the multi-objective vector $\mathcal{F}^i$. Next, the population $P_g$ is ranked via non-dominated sorting based on their performance across all objectives, yielding multiple Pareto fronts. To preserve solution diversity, crowding distances are computed within each front, and a crowded-comparison operator is used to select parent individuals for reproduction. Offspring population $Q_g$ is generated through crossover and mutation operations, and then merged with the current population $P_g$ to form the next generation $P_{g+1}$. After $N_g$ generations, the algorithm concludes with a final population $P_{N_g}$ that approximates the Pareto front of the defined multi-objective problem. This set of non-dominated solutions characterizes the trade-offs among competing loss terms and provides a principled basis for adaptively determining loss weights in PINN training. Further discussion on Pareto-front search is provided in Appendix A.1.

### 3.3 PARETO-GUIDED WEIGHTING METHOD

Upon constructing the Pareto front through the NSGA-II algorithm, the critical step is assigning appropriate loss weights to each objective in the PINN loss formulation. In this work, we develop the Pareto-guided weighting (PGW) method, which integrates performance-based feedback and variability cues from the current Pareto front to produce informed and adaptive loss weights. The Pareto-guided weighting method approach is grounded in two key observations: (1) different regions exhibit varying degrees of progress during training, and (2) the spread of each objective across the Pareto front reflects its underlying sensitivity and potential for improvement.

The key steps of Pareto-guided weighting method are shown in Algorithm 2. First, we evaluate the dispersion of each objective across the Pareto front, serving as a proxy for its influence on trade-off behavior. Let $\mathcal{F}_{ij} \in \mathbb{R}^{N_P \times N_o}$ denote the Pareto-optimal objective matrix consisting of $N_p$ solutions and $N_o$ objectives (including regional residuals, initial/boundary conditions, and data loss). To ensure comparability across different scales, the objective matrix $\mathcal{F}_{ij}$ are normalized via min-max scaling:

$$\mathcal{F}_{ij}^{norm} = \frac{\mathcal{F}_{ij} - \min_{i \in N_p} \mathcal{F}_{ij}}{\max_{i \in N_p} \mathcal{F}_{ij} - \min_{i \in N_p} \mathcal{F}_{ij}} \tag{6}$$

The spread of each normalized objective $j$ over the Pareto front is evaluated by its standard deviation:

$$\sigma_j = \sqrt{\frac{\sum_{i=1}^{N_p} \left( \mathcal{F}_{ij}^{norm} - \overline{F}_j^{norm} \right)^2}{N_p}} \tag{7}$$

where $\overline{F}_j^{norm} = \frac{1}{N_p} \sum_{i=1}^{N_p} \overline{F}_{ij}^{norm}$ is the mean values of $j$th normalized objective. This variability score captures how widely an objective's values fluctuate across the Pareto front. A higher $\sigma_j$ indicates that the objective is more sensitive to parameter changes, and may benefit from focused optimization.

The next stage aims to measure the progress of each loss term by comparing its current performance on the Pareto front with its historical baseline. For each objective $j$, the central tendency across the

Pareto front is defined as $\overline{\mathcal{F}}_j = \frac{1}{N_p} \sum_{i=1}^{N_p} \mathcal{F}_{ij}$. To assess training progress, we extract a historical baseline $\mathcal{L}_j^{pre}$ for each objective $j$ from the previous $w$ training steps:

$$\mathcal{L}_j^{pre} = \min_{t \in T} \mathcal{L}_{t,j}^{hist}, \quad \text{where } T = \{t \mid t = T_{\text{end}} - w, \ldots, T_{\text{end}}\} \tag{8}$$

where $\mathcal{L}_{t,j}^{hist}$ is the value of the $j$th loss term recorded at training iteration $t$, and $T$ represents a historical window of length $w$, ending at the current iteration $T_{\text{end}}$. This baseline mitigates the impact of stochastic fluctuations that are common in PINN training, where loss values can exhibit non-monotonic behavior due to the complex interplay between different regions $\Omega_m$. By selecting the minimum loss over the window, we ensure a robust and stable reference that captures the best recent performance of each objective. The relative improvement ratio is then defined as the ratio between its current average performance on the Pareto front and its historical baseline ($r_j = \frac{\overline{\mathcal{F}}_j}{\mathcal{L}_j^{\text{pre}}}$). A lower value of $r_j$ indicates that the current Pareto-optimal solutions exhibit significant improvement over past training steps, implying that the corresponding objective still holds potential for further improvement and may benefit from continued focus. In contrast, a higher value of $r_j$ suggests that recent progress has stagnated, and the objective may have reached a plateau, potentially requiring less emphasis in subsequent training iterations. To synthesize the effects of both progress and variability, a saliency score for each objective is defined and converted to normalized weights using a temperature-controlled softmax:

$$s_j = \frac{\sigma_j}{r_j}, \quad \alpha_j = \frac{N_o \cdot \exp(s_j/\mathcal{T}_j)}{\sum_{j=1}^{N_o} \exp(s_j/\mathcal{T}_j)} \tag{9}$$

where the saliency score $s_j$ serves as a composite indicator that captures both the variability of each objective and its recent improvement behavior, thus supporting a dynamic ranking of different regions in response to changing training dynamics and uncertainties. A softmax function with temperature scaling is applied to translate these scores into adaptive training weights for the PINN. The temperature parameter $\mathcal{T}_j$ modulates the sharpness of the softmax distribution. A comprehensive discussion of $\mathcal{T}_j$ can be found in the Appendix A.2.

### 3.4 WEIGHT-INFORMED REGIONAL RESAMPLING

In this subsection, we propose a resampling strategy that adaptively reallocates collocation points across regions based on their current learning importance. Specifically, once adaptive weights $\alpha_m$ are computed through the PGW method, these weights are not only used for loss balancing during backpropagation, but are also transformed into sampling probabilities with a tunable scaling factor $\gamma$:

$$\rho_m = \frac{\alpha_m^\gamma}{\sum_{m'=1}^M \alpha_{m'}^\gamma} \tag{10}$$

where $\rho_m$ denotes the sampling probability assigned to region $\Omega_m$. Intuitively, a higher sampling probability $\rho_m$ encourages the network to allocate more learning resources to challenging areas. Additionally, the adaptive weights $\alpha_m$ often exhibit small variations between different regions, since overly aggressive fluctuations can compromise stability. In such cases, directly converting $\alpha_m$ into probabilities may not sufficiently differentiate between regions of distinct learning importance. Therefore, the scaling factor $\gamma$ is introduced here to increase the sensitivity of probabilities to weight differences. Based on the sampling probabilities, the number of collocation points to be drawn from each region is determined as follows:

$$N_m = N_r \cdot \rho_m \tag{11}$$

where $N_r$ is the total number of residual collocation points used in the training phase, and $N_m$ is rounded to the nearest integer to ensure integer counts for sampling. For each region $\Omega_m$, a new subset of collocation points $\{(x_r^{j,m}, t_r^{j,m})\}_{j=1}^{N_m}$ is then randomly sampled from the spatio-temporal domain of $\Omega_m$. Once all $M$ subsets are resampled, they are merged into a new collocation pool $\mathcal{X}_r^{\text{new}}$.

The overall Pareto-guided regional sampling (PaRS) framework for PINN is finally presented in Algorithm 3. In the context of PINNs, we decouple the neural network parameter optimization from the multi-objective balancing task. Instead of using NSGA-II to update the neural network parameters $\theta$, the algorithm is applied to explore appropriate combinations of loss term weights that yield optimal trade-offs among the residual subdomain objectives and other losses. This provides a valuable decision-making tool for engineers, enabling them to tailor weight assignment strategies that better reflect the specific priorities and constraints of real-world applications.

---

**Algorithm 3:** Pareto-guided regional sampling (PaRS) framework

---

**Input:** Initial $\mathcal{X}_r$, number of regions $M$, resampling interval $N_{\text{inner}}$, total training epochs $E$
Apply K-Means to partition $\mathcal{X}_r$ into $M$ subdomains $\{\Omega_m\}_{m=1}^M$
**for** $e = 1$ **to** $E$ **do**
    **if** $e \bmod N_{inner} = 0$ **then**
        Approximate Pareto front: $\mathcal{F} \leftarrow Algorithm\ 1(\theta, k)$
        Apply Pareto-guided weighting method to derive weights:
          $\alpha_m, \alpha_b, \alpha_i, \alpha_d \leftarrow Algorithm\ 2(F, \mathcal{L}^{hist})$
        Derive sampling probabilities for each region: $\rho_m \leftarrow \alpha_m$
        Resample $N_m$ collocation points in each region
        Update collocation pool: $\mathcal{X}_r^{\text{new}} \leftarrow \bigcup_{m=1}^M \mathcal{X}_r^{(m)}$
    **end**
    Update model parameters $(\theta, k)$ via gradient descent on the composite loss of Eq. 4
**end**
**Output:** Trained model parameters $(\theta, k)$

---

The PGW method dynamically guides the training toward promising yet under-optimized regions of the objective space, which is especially valuable in inverse problems, where the parameter space is highly non-convex. These weights are then translated into sampling probabilities, guiding the dynamic resampling of collocation points from each subdomain. The updated residual point set $\mathcal{X}_r^{\text{new}}$ is then held fixed for the subsequent $N_{\text{inner}}$ training iterations, where $N_{\text{inner}}$ is a user-defined hyperparameter controlling the frequency of resampling. Periodically updating the collocation set ensures that the network progressively adapts to evolving error patterns and focuses on the most informative regions of the domain.

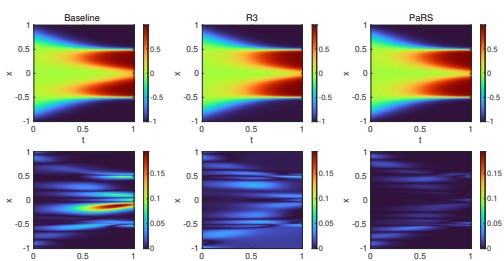

Figure 2: Allen-Cahn equation: Predicted $u(x, t)$ solutions (top) and absolute errors (bottom) using different methods.

## 4 RESULTS AND DISCUSSION

In this section, we evaluate the performance of the proposed PaRS framework through the inverse problem of both PDE and ODE systems: Allen-Cahn equation, convection equation, and a nonlinear chemical process. We also conduct a comprehensive comparison with two strategies: a baseline PINN trained with a fixed set of uniformly sampled collocation points and a state-of-the-art sampling strategy known as Retain-Resample-Release (R3) algorithm (Daw et al., 2023). The details of the experiment setups and hyperparameter settings are provided in Appendix B.

### 4.1 ALLEN-CAHN EQUATION

We consider the Allen-Cahn equation as follows:

$$\frac{\partial u}{\partial t} = D\frac{\partial^2 u}{\partial x^2} + 5\left(u - u^3\right), \quad x \in [-1, 1], t \in [0, 1],$$
$$u(x, 0) = x^2 \cos(\pi x), \quad u(-1, t) = u(1, t) = -1, \tag{12}$$

where the diffusion coefficient $D = 0.001$ will be used as an unknown parameter to estimate during the PINN training process. As shown in Fig. 2, the baseline PINN relying on uniformly sampled collocation points exhibits large localized errors around the steep gradient regions. The R3 method alleviates some of these issues by adaptively redistributing collocation points, leading to reduced errors compared to the baseline. In contrast, PaRS-PINN can accurately capture both the smooth variations and the sharp transition layers characteristic of the Allen–Cahn dynamics. As shown in Fig. 4a, all models exhibit a rapid drop in test loss during the initial training phase. After their

respective strategies are activated (around the 1000th epoch), PaRS-PINN clearly achieves the fastest convergence and the lowest final test loss. Fig. 4b shows that the PaRS method consistently provides the closest approximation to the true parameter value, reflecting its ability to balance competing losses and allocate sufficient learning resources to regions significant for parameter identification. Further details about the experimental results of Allen-Cahn equation are presented in Appendix B.1.

## 4.2 CONVECTION EQUATION

We consider a one-dimensional convection equation, which is a prototypical hyperbolic PDE widely used to model transport phenomena in engineering applications:

$$\frac{\partial u}{\partial t} + \beta \frac{\partial u}{\partial x} = 0, \quad x \in [0, 2\pi], t \in [0, 1] \tag{13}$$
$$u(x, 0) = \sin(x), \quad u(0, t) = u(2\pi, t)$$

where the convection coefficient $\beta = 50$ is the unknown parameter that will be learned during the training process of PINNs. Fig. 3 shows that the PaRS-PINN model achieves superior accuracy and produces predictions that are in line with the reference solution. Similarly, as shown in Fig. 4c and Fig. 4d, the PaRS-PINN model exhibits a superior performance for convergence speed and parameter identification among all models, which converges faster to the true parameter values. Further details about the experimental results of the convection equation are presented in Appendix B.2.

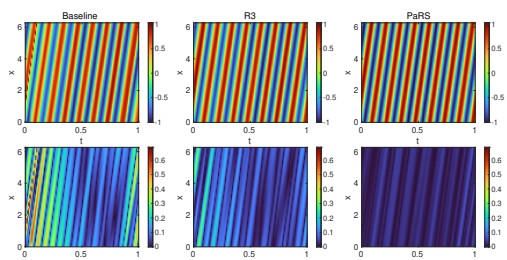

Figure 3: Convection equation: Predicted $u(x, t)$ solutions (top) and absolute errors (bottom) using different models.

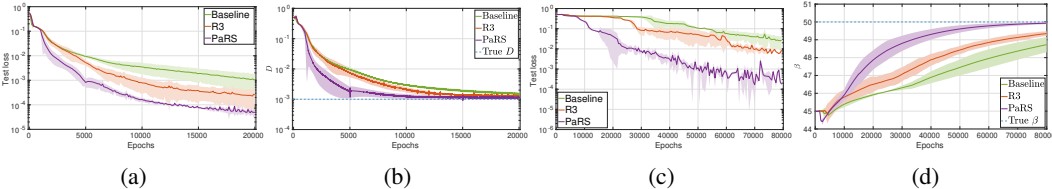

|       |       |       |       |
|-------|-------|-------|-------|
| (a)   | (b)   | (c)   | (d)   |

Figure 4: Evolution of test loss and parameter value of Allen–Cahn equation and convection equations: (a) and (b) show the Allen–Cahn equation results, including the evolution of the test loss and the approximation of the true parameter value $D$. (c) and (d) show the convection equation results, including the evolution of the test loss and the approximation of the true parameter value $\beta$.

## 4.3 APPLICATION TO A NONLINEAR CHEMICAL PROCESS

For process modeling, the PINN framework is often extended to physics-informed recurrent neural networks (PIRNNs), which are well-suited for handling time-series data due to their ability to incorporate temporal dependencies while simultaneously enforcing physical constraints. This section evaluates the proposed PaRS framework using a continuous stirred tank reactor (CSTR) as a representative benchmark system. The dynamics of the CSTR can be expressed as follows (Heidarinejad et al., 2012):

$$\frac{dC_A}{dt} = \frac{F_{in}}{V}(C_{A0} - C_A) - k_0 e^{\frac{-E}{R_c T}} C_A^2, \quad \frac{dT}{dt} = \frac{F_{in}}{V}(T_0 - T) + \frac{-\Delta H}{\rho_L C_p} k_0 e^{\frac{-E}{R_c T}} C_A^2 + \frac{Q_h}{\rho_L C_p V} \tag{14}$$

where $C_A$ and $T$ are two state variables that represent the concentration and temperature of the system, respectively. The objective is to train a PIRNN model to accurately learn the dynamics of CSTR for downstream tasks such as prediction and control. The feed flow rate $F_{in}$ and kinetic parameter $k_0$ are treated as unknown parameters to be identified during training. The state vector and control inputs of CSTR are defined as $x^T = [C_A - C_{As} \quad T - T_s]$ and $u^T = [\Delta C_{A0} \quad \Delta Q_h]$, where $(C_{As}, T_s)$ is the steady-state.

Table 1: Comparison of loss errors $e_{loss}$ and $L^2$ relative errors $e_{L_2}$ for three experiments under different numbers of collocation points $N_r$.

| | Method | $N_r = 500$ | | $N_r = 1000$ | | $N_r = 2000$ | |
|---|---|---|---|---|---|---|---|
| | | $e_{loss}$ | $e_{L_2}$ | $e_{loss}$ | $e_{L_2}$ | $e_{loss}$ | $e_{L_2}$ |
| **Allen–Cahn** | Baseline | 7.24e-4 | 3.80e-2 | 1.08e-3 | 4.39e-2 | 2.61e-4 | 2.28e-2 |
| | R3 | 1.81e-4 | 1.90e-2 | 2.55e-4 | 2.14e-2 | 1.23e-4 | 1.57e-2 |
| | PaRS | 4.80e-5 | 9.79e-3 | 4.91e-5 | 9.82e-3 | 4.75e-5 | 9.74e-3 |
| | Method | $N_r = 500$ | | $N_r = 1000$ | | $N_r = 2000$ | |
| | | $e_{loss}$ | $e_{L_2}$ | $e_{loss}$ | $e_{L_2}$ | $e_{loss}$ | $e_{L_2}$ |
| **Convection** | Baseline | 3.00e-2 | 2.45e-1 | 1.53e-2 | 1.75e-1 | 2.45e-2 | 2.21e-1 |
| | R3 | 5.06e-3 | 1.01e-1 | 7.19e-3 | 1.20e-1 | 1.35e-3 | 5.19e-2 |
| | PaRS | 2.12e-4 | 2.07e-2 | 3.10e-4 | 2.41e-2 | 5.89e-4 | 3.43e-2 |
| | Method | $N_r = 100$ | | $N_r = 400$ | | $N_r = 800$ | |
| | | $e_{loss}$ | $e_{L_2}$ | $e_{loss}$ | $e_{L_2}$ | $e_{loss}$ | $e_{L_2}$ |
| **CSTR** | Baseline | 5.88e-5 | 7.67e-3 | 2.88e-5 | 5.35e-3 | 5.39e-5 | 7.34e-3 |
| | R3 | 3.09e-5 | 5.56e-3 | 2.66e-5 | 5.13e-3 | 2.10e-5 | 4.59e-3 |
| | PaRS | 1.35e-5 | 3.67e-3 | 9.29e-6 | 3.03e-3 | 1.16e-5 | 3.41e-3 |

As shown in Fig. 5a, the state trajectories predicted by the PaRS-PIRNN model closely match the ground-truth response of the CSTR, which indicates that PaRS-PIRNN model can reconstruct the nonlinear dynamics of the reactor. Fig. 5b further demonstrates the evolution of the test loss during training. As shown in Fig. 5c, the task of parameter identification is particularly difficult in this case study due to the magnitude of $k_0$, which is on the order of $10^6$. Even minor deviations in estimating this parameter can significantly impact the reaction rate and destabilize model training. Despite this, the PaRS-PIRNN exhibits the most rapid and precise convergence for both unknown parameters, reaching values close to the true ones by around epoch 2000 and remaining stable thereafter. Additionally, Table 1 compares the accuracy of different methods for three case studies with varying numbers of collocation points. The proposed PaRS method consistently achieves the lowest errors across various sampling points, demonstrating its robustness in handling the nonlinear dynamics of PDEs and chemical process.

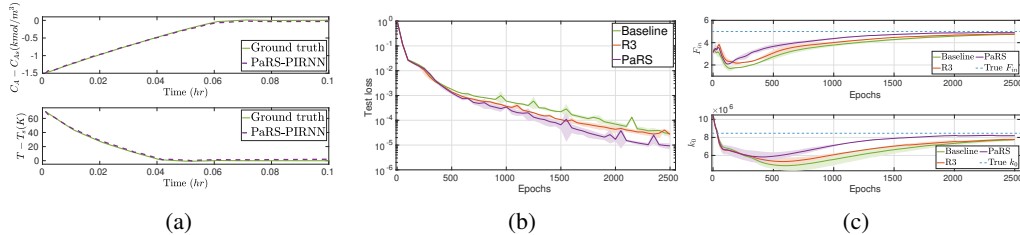

| (a) | (b) | (c) |

Figure 5: CSTR: (a) The state trajectories predicted by PaRS-PIRNN. (b) Evolution of the test loss. (c) Approximation of the true parameter value $F_{in}$ and $k_0$.

## 5 CONCLUSION

In this work, we developed a PaRS framework for jointly resampling and tuning the weight of collocation points in physics-informed neural networks. Specifically, the residual loss is first decomposed into subregional objectives. An MOO algorithm is performed to search the Pareto front that provides informative trade-offs among regional residuals and other losses. Subsequently, a Pareto-guided weighting method is developed to provide adaptive weight for each objective, while the weight-informed resampling mechanism is then developed to increase collocation density in error-prone regions. Extensive experiments demonstrated that the proposed PaRS framework achieves faster convergence, improved predictive accuracy, and more reliable parameter identification compared with both baseline PINNs and the state-of-the-art R3 strategy.

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

# A  FURTHER DISCUSSIONS ON PARS

## A.1  DISCUSSION ON PARETO-FRONT SEARCH

The regional residuals $\mathcal{L}_r^{(m)}$ often reflect distinct dynamic regimes or approximation difficulties across the domain. In many physics-informed problems, the solution may exhibit steep gradients, boundary layers, or rapidly changing dynamics that are highly localized in certain spatial or temporal regions. Therefore, collocation points from different subdomains may interact or compete in complex ways during training. For example, the network's effort to reduce the residual in one region may inadvertently increase the error in another. This competitive behavior among regions makes it extremely challenging to manually design loss weights and sampling strategies that balance the learning across all regions effectively. In PaRS framework, the residual loss from each spatial region $\mathcal{L}_r^{(m)}$ is treated as an independent objective, allowing for a finer-grained handling of the internal structure of the residual error. In addition, the initial condition loss $\mathcal{L}_{IC}$, boundary condition loss $\mathcal{L}_{BC}$, and data loss $\mathcal{L}_{data}$ are also incorporated as separate objectives.

## A.2  DISCUSSION ON PARETO-GUIDED WEIGHTING METHOD

Multi-objective optimization (MOO) is widely employed to manage competing objectives in engineering design and decision-making (Sharma & Kumar, 2022; Gunantara, 2018). Many algorithms have been developed to approximate Pareto fronts in nonconvex landscapes (Ye et al., 2022; Mirjalili et al., 2017). Points on a Pareto front encode trade-offs among objectives rather than a single prioritized choice. Therefore, a weighting scheme is required to select a final design that aligns with practical needs or domain-specific preferences (Ye et al., 2025). Common approaches include criteria-importance techniques such as CRITIC (Diakoulaki, Mavrotas, and Papayannakis 1995), entropy-based weighting (Li et al., 2014), and variance-driven statistical methods (Yusop et al., 2015), which assign relative importance to individual objectives. The other two popular subjective methods are stepwise weight assessment ratio analysis (Keršuliene et al., 2010) and best-worst method (Rezaei, 2015).

In PaRS framework, a lower $\mathcal{T}_j$ intensifies focus on objectives with high $s_j$, suitable during early training phases when model convergence is still ongoing. A higher $\mathcal{T}_j$ flattens the weight distribution, aiding stable fine-tuning during later stages. To mitigate the limitations of fixed weighting, especially the risk of overemphasizing physics-based terms before the parameters are sufficiently identified, PGW adjusts $\mathcal{T}_j$ adaptively according to the current value of the data-driven loss. Specifically, if $\mathcal{L}_{\text{data}}$ remains above a defined threshold $\epsilon$, suggesting insufficient parameter identification, a lower temperature (e.g., $\mathcal{T}_{\text{data}} = 0.5$) is assigned to increase its influence. Once $\mathcal{L}_{\text{data}} < \epsilon$, the temperatures for all objectives are unified (e.g., $\mathcal{T}_j = 1$) to allow balanced training and prevent overfitting to any single objective.

## A.3  DISCUSSION ON REGIONAL RESAMPLING

In traditional PINN frameworks, the set of residual collocation points is typically fixed throughout training and uniformly sampled across the entire spatiotemporal domain. However, such a static sampling strategy fails to reflect the evolving difficulty of different regions as training progresses. Certain subdomains may initially exhibit large approximation errors, demanding more collocation points for accurate supervision, while others may converge early and no longer require extensive sampling. Continuing to treat all regions equally wastes computational resources and hinders the model to focus on underperforming areas. Compared to static or uniform sampling strategies, the PaRS method enables a more efficient training process. It reduces redundancy in overfitted areas while systematically increasing the sampling density in underexplored or error-prone regions. This synergy between loss weighting and dynamic resampling allows the PaRS-PINN to achieve faster convergence, especially in problems with heterogeneous dynamics or localized complexities.

## B EXPERIMENTAL DETAILS

Our goal is to simultaneously learn the solution and system parameters at certain times by using different PINN model with limited data. The performance of different methods is assessed based on several metrics including predictive accuracy, convergence speed, and quality of parameter identification. Due to the inherent randomness in PINN results arising from random sampling, neural network initialization, and optimizer dynamics, each experiment is repeated at least five times for every case study.

### B.1 FURTHER DETAILS ABOUT ALLEN-CAHN EQUATION

The Allen–Cahn equation is a prototypical reaction–diffusion model originating from phase-field theories of microstructure evolution. It governs the dynamics of an order parameter $u(x,t) \in [-1, 1]$ that distinguishes coexisting phases and evolves under the competition between interfacial smoothing diffusion and a double-well reaction potential. Due to its simplicity and rich dynamics, the Allen–Cahn model is widely used in materials science (e.g., phase separation and grain growth), combustion, image processing, and as a benchmark for stiff nonlinear PDE solvers. Despite its relative simplicity in form, solving the Allen–Cahn equation poses significant challenges due to its nonlinear reaction term and the sharp interface dynamics that emerge during evolution. In particular, the solution often exhibits steep gradients and localized transition layers, which require dense collocation point coverage to be resolved accurately. Uniform sampling strategies may oversample smooth regions and undersample interface zones, leading to inefficient training and poor accuracy.

The reference solutions used in this experiment are obtained from Lu et al. (2021), which provides a numerical solution $u(x,t)$ of the Allen-Cahn equation computed over a grid of $201 \times 101$ points, resulting in a total of 20,301 reference data points. The training dataset is comprised of $N_r = 1000$ collocation points, $N_i = 400$ initial condition points, $N_b = 400$ boundary condition points, and $N_{data} = 100$ data points randomly selected from the available observations. The dataset consisting of all 20,301 points is used as a test set to evaluate the predictive accuracy of the trained PINN models via MSE and $L^2$ relative errors. All the PINN models in this experiment share the same network architecture, which consists of 3 hidden layers, each with 40 neurons, and tanh activation functions are employed throughout. The unknown diffusion coefficient $D$ is initialized at $0.5$, while all the initial loss weights $\alpha_m, \alpha_i, \alpha_b, \alpha_d$ are set to 1 at the start of training. Each model is trained for a total of 20,000 epochs using the Adam optimizer with a fixed learning rate of $0.001$. For the proposed PaRS-PINN, we set $N_p = 50$, $N_g = 100$, $w = 100$, $\epsilon = 0.001$, $M = 3$ and $N_{inner} = 5000$.

As shown in Fig. 4a, the test loss of PaRS quickly drops to around $10^{-3}$ in only about 5,000 epochs, and subsequently converges to $10^{-4}$ ahead of other methods. Considering the current amount of training data and the number of collocation points, this level of accuracy (around $10^{-4}$) can be considered close to the practical limit of the system. Compared with the baseline and R3 methods, it not only reduces errors more effectively but also shows more stable performance across different runs.

Fig. 6 further validates the accuracy of the proposed PaRS framework by comparing the evolution of the solution at selected time snapshots ($t = 0.25, 0.5, 0.75$). In particular, even as the system evolves and sharp transition layers emerge around $t = 0.50$ and $t = 0.75$, the predicted profiles remain overlapped with the exact solution. Fig. 7 shows the logarithmic evolution of the different loss components and the corresponding adaptive weight $\alpha$ during the training process of the Allen–Cahn equation using PaRS. It can be observed that the data loss weight $\alpha_d$ is temporarily emphasized in the early epochs to ensure reliable parameter identification, while the boundary condition weight $\alpha_b$ gradually increases later to enforce global consistency. The regional residual weights $\alpha_1, \alpha_2, \alpha_3$ remain balanced without collapsing, ensuring that all subdomains receive adequate attention. This dynamic adjustment allows the model to allocate computational resources adaptively, focusing on the objectives that are the most critical at each stage of training.

### B.2 FURTHER DETAILS ABOUT CONVECTION EQUATION

The convection equation is one of the fundamental PDEs in fluid dynamics and transport phenomena. It describes the propagation of a conserved scalar quantity (such as heat, mass, or concentration) carried by a constant velocity field without diffusion or reaction effects. Unlike parabolic PDEs,

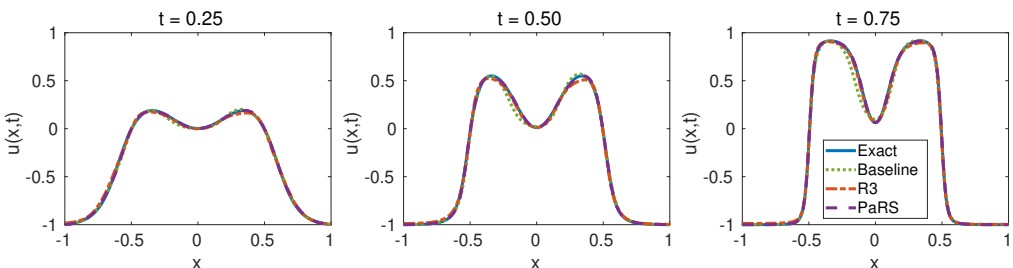

Figure 6: Allen-Cahn equation: The reference solution and predictions of different models at different time points.

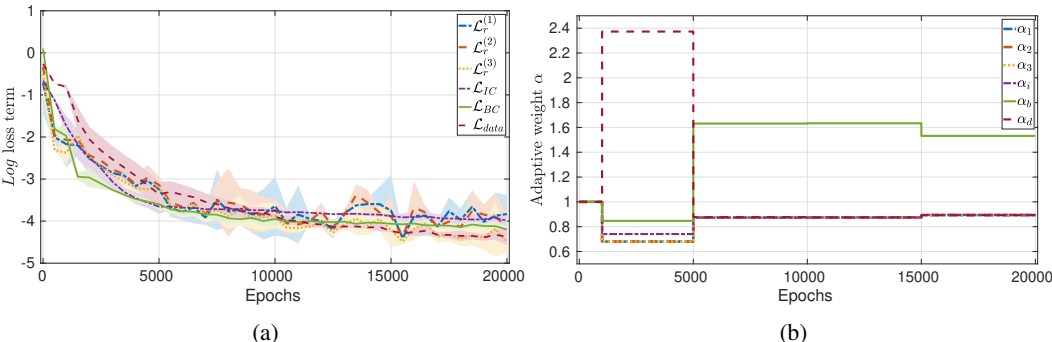

Figure 7: Allen-Cahn equation: (a) Evolution of the residual loss $\mathcal{L}_r^{(M)}$, initial condition loss $\mathcal{L}_{IC}$, boundary condition loss $\mathcal{L}_{BC}$, and data loss $\mathcal{L}_{data}$ during the training process. (b) Evolution of the adaptive weights $\alpha$ during the training process for PaRS-PINN model.

hyperbolic equations propagate information along characteristic lines with minimal diffusion, which leads to solutions that preserve sharp features over time. Uniform collocation sampling often fails to capture these characteristics effectively, resulting in numerical dissipation or phase errors in the predicted waveforms.

For this problem, the reference solution is obtained analytically by using the Fourier transform (Krishnapriyan et al., 2021). We take $N_r = 1000$, $N_i = 256$, $N_b = 100$ and $N_{data} = 25$. The predictive accuracy of the PINN models is evaluated using the MSE and the relative $L_2$ error computed on a dense test set containing 25,600 extra points. All models are configured with three hidden layers of 40 neurons each and employ the tanh activation function. The unknown convection coefficient $\beta$ is initialized at 45, and the initial loss weights $\alpha_m, \alpha_i, \alpha_b, \alpha_d$ are set to 1 at the beginning of the training. The optimization is carried out using the Adam optimizer with a learning rate of 0.001 for a total of 80,000 epochs. For the proposed PaRS-PINN, the parameters are chosen as $N_p = 50$ $N_g = 100$, $w = 100$, $\epsilon = 0.001$, $M = 3$, and $N_{\text{inner}} = 5000$.

As shown in Fig. 3, the baseline PINN shows clear difficulties in learning sharp features, leading to significant phase shifts and accumulated errors along the spatiotemporal domain. The R3 method improves the accuracy by dynamically refining regions with large residuals. Fig. 4c illustrates the evolution of the test loss during the training process for different PINN models. The baseline PINN exhibits a relatively slow decrease in test loss and eventually stabilizes at a higher error level, reflecting the limitations of using a fixed sampling strategy for hyperbolic PDEs. In contrast, the R3 method begins to outperform the baseline after around 20,000 epochs, demonstrating the benefit of dynamically updating collocation points based on residual information. In particular, the PaRS method shows a clear acceleration in convergence once its adaptive mechanism is activated, achieving substantially lower test losses compared to both the baseline and R3 models.

Fig. 8 shows the logarithmic evolution of the different loss components and the corresponding adaptive weight $\alpha$ for the convection equation using PaRS. At the early stage of training, the data loss $\mathcal{L}_{data}$ and boundary loss $\mathcal{L}_{BC}$ are larger, leading to an increase in their associated weights. This adjustment prevents the model from prematurely overemphasizing the physics-driven residuals

before the unknown parameters and system dynamics are reasonably captured. Once the unknown parameters and system dynamics are roughly captured, the framework quickly redistributes the weights from the data-driven terms $\alpha_d$ to the physics-driven terms $\mathcal{L}_r^{(M)}$, achieving faster and more stable convergence. In this case study, the PaRS framework is activated only during the first 40,000 epochs, which is sufficient to capture the critical training dynamics and demonstrate its effectiveness in guiding parameter identification. These results highlight the potential of PaRS framework to serve as a general framework for solving inverse problems of PDEs with complex dynamics and uncertain parameters.

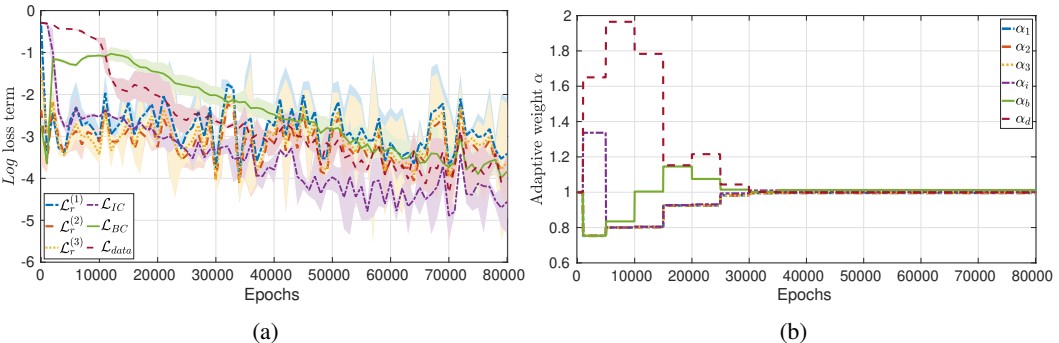

(a) (b)

Figure 8: Convection equation: (a) Evolution of the residual loss $\mathcal{L}_r^{(M)}$, initial condition loss $\mathcal{L}_{IC}$, boundary condition loss $\mathcal{L}_{BC}$, and data loss $\mathcal{L}_{data}$ during the training process. (b) Evolution of the adaptive weights $\alpha$ during the training process for PaRS-PINN model.

### B.3 FURTHER DETAILS ABOUT CSTR

Physics-informed learning techniques have attracted significant attention in industrial process modeling due to their ability to incorporate domain knowledge into black-box models. In this work, we consider a well-stirred, non-isothermal CSTR that performs an irreversible and exothermic second-order chemical reaction, $A \rightarrow B$. The conversion rate of reactant $A$ to product $B$ is governed by the kinetics expression $r_B = k_0 e^{-E/(R_c T)} C_A^2$, where $k_0$, $E$ and $R_c$ represent the pre-exponential factor, activation energy, and ideal gas constant, respectively. Thermal energy is supplied to or removed from the reactor through an external jacket that operates at a heat exchange rate of $Q_h$. The reactor vessel holds a liquid of volume $V$, assumed to have a constant density $\rho_L$ and a specific heat capacity $C_p$. The term $\Delta H$ denotes the enthalpy change associated with the reaction. The inlet conditions are specified by the feed temperature $T_0$ and the volumetric flow rate $F_{in}$. A complete list of system parameters is summarized in Table 2. To generate observational data for training and evaluation, we solve the governing equations of Eq. 14 using the explicit Euler method with an integration time step of $h_c = 10^{-3}$ hr. The unknown parameters are initialized as $F_{in} = 3.125$ and $k_0 = 1.057 \times 10^7$.

The PIRNN model is designed to predict the future state trajectory of CSTR over a 10-step prediction horizon, based on the current state vector $x^T$ and control inputs $u^T$. We set $N_r = 400$ and $N_{data} = 50$. To ensure a fair performance comparison between different models, the predictive accuracy is evaluated using a separate set of 11,800 samples during training. All PIRNN models used in this case study are configured with two hidden layers, comprising 128 units and 64 recurrent units, respectively. The initial loss weights $\alpha_m$ and $\alpha_d$ are set to 1. Model training is carried out using the Adam optimizer with a learning rate of 0.001, and all models are trained for a total of 2,500 epochs. For the PaRS-PIRNN, we set $N_p = 50$, $N_g = 30$, $w = 10$, $\epsilon = 0.0001$, $M = 3$ and $N_{inner} = 300$.

As shown in Fig. 5b, although the baseline PIRNN converges slowly and R3 provides moderate improvements, the PaRS-PIRNN consistently achieves the lowest test loss and exhibits faster convergence. Fig. 9 shows the evolution of individual loss terms and adaptive weights. Specifically, PaRS-PIRNN places more emphasis on data loss in the early stages of training to determine uncertain parameters. Once the system parameters are approximately identified, the weights are gradually shifted toward physics-driven residuals, ensuring that physical constraints are more strongly enforced in later stages. At the same time, the residual losses are decomposed region-wise, allowing the framework to assign weights and resampling at a finer granularity and thereby improve the performance of the PIRNN model in locally challenging regions. These results demonstrate the

effectiveness of the proposed PaRS framework in balancing regional loss, exploiting limited data, and accurately estimating uncertain parameters, thereby providing a reliable solution for challenging inverse problems in industrial process modeling.

Table 2: Process parameters of the CSTR

| | |
|---|---|
| $E = 5 \times 10^4$ kJ/kmol | $C_{A0_s} = 4$ kmol/m$^3$ |
| $V = 1$ m$^3$ | $C_{As} = 1.95$ kmol/m$^3$ |
| $F_{in} = 5$ m$^3$/hr | $Q_s = 0.0$ kJ/hr |
| $R_c = 8.314$ kJ/kmol $\cdot$ K | $T_s = 402$ K |
| $T_0 = 300$ K | $\Delta H = -1.15 \times 10^4$ kJ/kmol |
| $C_p = 0.231$ kJ/kg $\cdot$ K | $\rho_L = 1000$ kg/m$^3$ |
| $k_0 = 8.46 \times 10^6$ m$^3$/kmol $\cdot$ hr | |

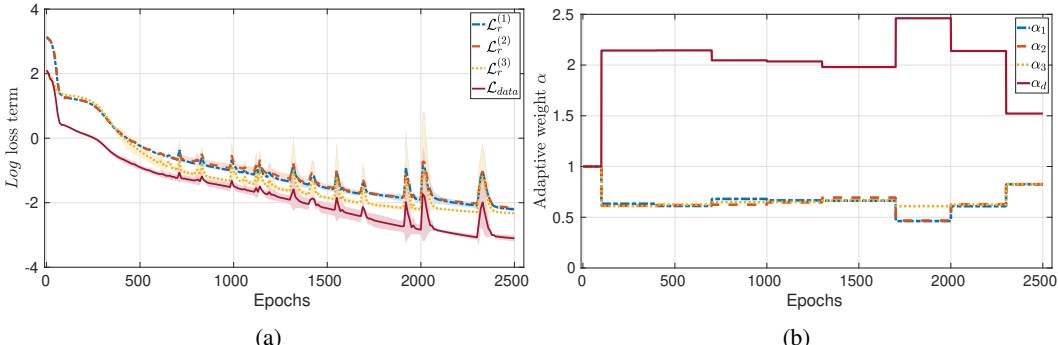

(a)            (b)

Figure 9: CSTR: (a) Evolution of the residual loss $\mathcal{L}_r^{(M)}$ and data loss $\mathcal{L}_{data}$ during the training process. (b) Evolution of the adaptive weights $\alpha$ during the training process for PaRS-PIRNN model.

## C  COMPUTATIONAL RESOURCES

All experiments are conducted on an Ubuntu 20.04.4 LTS server running Python 3.8.10 and PyTorch 1.11.0, equipped with an 11th-Gen Intel Core i7-11700 CPU (16 threads, 2.5 GHz base clock), 32 GB RAM, integrated Intel UHD Graphics (Rocket Lake GT1), and 1.5 TB of local storage.

Table 3 reports the computational times for the baseline and the proposed PaRS framework for three case studies. Compared with the baseline method, the integration of the PaRS framework into training incurs a slight increase in computational time, but the overhead remains moderate and controllable. In practice, each PaRS activation constitutes only a small fraction of the total training time (e.g., the convection equation requires roughly 11 s), indicating that the observed gains in accuracy and convergence are achieved with little additional cost. Furthermore, the Pareto-front search phase is readily parallelizable on distributed hardware, enabling simultaneous evaluation of candidate solutions and weight updates across multiple nodes or GPUs and thereby substantially reducing the computational time of each update.

Table 3: Computational time for training different case studies.

| Case study | Baseline | PaRS | One-time PaRS |
|---|---|---|---|
| Allen-Cahn equation | 5 min 10 s | 5 min 59 s | 12 s |
| Convection equation | 23 min 78 s | 25 min 55 s | 11 s |
| CSTR | 41 min 25 s | 46 min 15 s | 40 s |

## D  REPRODUCIBILITY STATEMENT

We provide the detailed experimental settings for each case study in Appendix B. In addition, the source code has been submitted as supplementary material to facilitate reproducibility.

## E  USE OF LARGE LANGUAGE MODELS

Large language models (LLMs) were only used for improving the writing clarity and grammar of this manuscript. No LLMs were involved in developing algorithms, conducting experiments, or analyzing results.

