# OpenReview forum: "Pareto-Guided Regional Sampling for Adaptive Collocation in Physics-Informed Neural Networks"
_ICLR.cc/2026/Conference — ICLR 2026 Conference Withdrawn Submission_

### Official Review · Reviewer_wbrr · 2025-10-15

**Soundness:** 2
**Presentation:** 2
**Contribution:** 2
**Rating:** 2
**Confidence:** 4

**Summary:**

This paper introduces a framework for collocation sampling and adaptive loss weighting called Pareto-Guided Regional Sampling (PaRS). PaRS decomposes the domain using a k-means clustering algorithm, then computes Pareto-guided weights via a Pareto-front search with NSGA-II. Regional sampling probabilities are derived from these weights, and PaRS applies the resulting loss weights and sampling probabilities to each subdomain. In experiments, the proposed method outperforms the baseline and the R3 sampling method on the Allen–Cahn equation, the convection equation, and the CSTR system.

**Strengths:**

**Strengths**

- The idea of adaptively adjusting both loss weights and the number of sampled collocation points per subdomain based on Pareto front search seems novel.

- The proposed consistently outperforms the baseline approaches across benchmark PDEs.

**Weaknesses:**

**Weaknesses**
- The explanation and analysis of the proposed methodology are insufficient. PaRS reuses the weights $\alpha_m$ computed through Pareto front search, as both subdomain-specific loss weights and sampling allocation ratios. Since applying distinct loss weights across subdomains has an effect analogous to varying the number of sampled points per subdomain, this structure effectively introduces double weighting across domains. However, the paper provides no analysis of this potential redundancy. For instance, an ablation study isolating the effects of adaptive loss weighting alone or enforcing uniform loss weights across subdomains would clarify the contribution of each component. In addition, PaRS employs K-means clustering for domain decomposition, but this makes the subdomain boundaries implicit. The paper does not explain how uniform resampling is achieved within each subdomain under this setup.
- The set of comparison baselines is limited. Since PaRS integrates both adaptive loss weighting and collocation sampling, it should be compared against representative methods from both categories—for example, adaptive sampling approaches such as RAD [1], RAR [2], or RoPINN [3], and adaptive weighting schemes such as LRA [4] or NTK-based weighting [5]. However, the paper only compares PaRS with a single baseline, R3, which provides an incomplete evaluation of its relative strengths.
- The experimental evaluation is insufficient to fully validate the proposed method. The paper tests PaRS on only three relatively simple benchmark equations. In the PINN literature, model performance is often sensitive to initialization and randomness, so reporting the mean and standard deviation over multiple random seeds is standard practice. However, the authors report only single-run results, limiting the statistical reliability of the conclusions.
- The analysis of computational cost is lacking. The proposed algorithm is computationally demanding, as its complexity scales with both the number of collocation points and the model size. Yet, the paper evaluates PaRS only on small-scale settings, up to 2000 collocation points and networks with three hidden layers of 40 units (for Allen-Cahn equation) . A more comprehensive analysis on larger models and denser collocation grids is needed to assess the scalability and efficiency of the approach.

[1] Nabian, M. A., Gladstone, R. J., and Meidani, H. Efficient training of physics-informed neural networks via importance sampling. Computer-Aided Civil and Infrastructure Engineering, 36(8):962–977, 2021.

[2] Wu, C., Zhu, M., Tan, Q., Kartha, Y., and Lu, L. A comprehensive study of non-adaptive and residual-based adaptive sampling for physics-informed neural networks. Computer Methods in Applied Mechanics and Engineering, 403:115671, 2023.

[3] WU, Haixu, et al. Ropinn: Region optimized physics-informed neural networks. Advances in Neural Information Processing Systems, 2024, 37: 110494-110532.

[4] Sifan Wang, Yujun Teng, and Paris Perdikaris. Understanding and mitigating gradient flow pathologies in physics-informed neural networks. SIAM Journal on Scientific Computing, 43(5):A3055–A3081, 2021.

[5] Sifan Wang, Xinling Yu, and Paris Perdikaris. When and why PINNs fail to train: A neural tangent kernel perspective. Journal of Computational Physics, 449:110768, 2022.

**Questions:**

**Questions**

1. Are the effects of adaptive loss weighting and subdomain-wise collocation point rebalancing independent of each other? (See Weakness 1)

2. Does the proposed method outperform various existing loss balancing techniques and sampling methods? (See Weakness 2)

3. Does the computational cost remain manageable when using a larger number of collocation points or larger network architectures? (See Weakness 4)

4. In my opinion, the number of subdomains $M$ plays a crucial role in the proposed method. However, the paper does not provide a sensitivity analysis with respect to $M$. How does the performance of the algorithm and its computational cost vary as $M$ changes?

5. Why is the residual loss weight of the baseline model set to 3 in the Allen–Cahn equation code provided in the supplementary material? Since no explanation is given for this choice, can the authors provide a clear rationale for it?

---

### Official Review · Reviewer_bVUv · 2025-10-16

**Soundness:** 2
**Presentation:** 3
**Contribution:** 2
**Rating:** 4
**Confidence:** 3

**Summary:**

This paper proposes a novel Pareto-guided regional sampling (PaRS) framework for adaptive collocation point sampling and tuning. The core of PaRS is a Pareto front based on domain decomposition to capture trade-offs among losses. Furthermore, an adaptive loss weighting and a dynamic resampling approach based on Pareto-guided weights are introduced. Experiments show that PaRS outperforms standard uniform sampling methods and SOTA adaptive collocation point methods.

**Strengths:**

* The motivation is clearly described. The authors also provide a thorough explanation of the challenges in loss weighting and sampling for training PINNs.
* Experiments on multiple PDE/ODE tasks demonstrate the effectiveness of PaRS.

**Weaknesses:**

* Figure 1 is too brief and lacks sufficient useful information. The reviewer suggests that a more detailed description of the Pareto front search and Pareto-guided weighting would be helpful. Additionally, the presentation of domain decomposition is inconsistent with the description. In Line 153, the authors claim to use K-means clustering for decomposition, but the schematic more closely resembles a uniform grids method.
* The paper repeatedly emphasizes the importance of domain decomposition for PaRS and provides extensive explanations (e.g., in Appendix A.1). However, it lacks sufficient ablation experiments to support this claim.
* The advantages of using K-means clustering for domain decomposition lack comparison (e.g., visualization or ablation study)
* The paper contains several subjective judgments. For instance, in Line 266: "A higher $\sigma_j$ indicates that the objective is more sensitive to parameter changes, and may benefit from focused optimization." The reviewer argues that for sensitive objectives, reducing the loss weights might facilitate a more stable training (Multi-Task Learning Using Uncertainty to Weigh Losses for Scene Geometry and Semantics, CVPR). It is recommended that the authors support such claims with references to relevant papers or conduct further analysis via ablation studies.
* PaRS involves a large number of hyperparameters, making it necessary to perform ablation studies on them.
* The computational complexity of PaRS is high, which may render the proposed method infeasible for larger-scale PDE solving problems.

**Questions:**

* To my best knowledge, NSGA-II is more suitable for low-dimensional problems, but both the number of objectives and the solution dimensionality in this work are very high. Could this lead to a severe curse of dimensionality, subsequently causing the Pareto front search in Section 3.2 to fail?

---

### Official Review · Reviewer_qEXe · 2025-10-28

**Soundness:** 2
**Presentation:** 2
**Contribution:** 2
**Rating:** 4
**Confidence:** 4

**Summary:**

The paper proposes a new framework called Pareto-guided Regional Sampling for achieving adaptive collocation point sampling and loss weighting in Physics-Informed Neural Networks. Its core idea is to divide the spatial domain into multiple sub-regions and treat the residual loss, initial/boundary condition loss, and data loss of each region as competing optimization objectives, thereby constructing a multi-objective optimization problem. The authors use the NSGA-II algorithm to approximate the Pareto front among these objectives and design a Pareto-guided weighting method based on this front. This method dynamically calculates the weight of each objective by combining the variability of the objectives on the Pareto front and their recent improvement relative to a historical baseline. These weights are further used to guide a region-level resampling strategy, increasing the density of collocation points in regions with larger errors or insufficient learning.

The authors validated the method on three inverse problems: the Allen-Cahn PDE, a hyperbolic PDE, and a CSTR ordinary differential equation system. Experimental results show that compared to standard PINN and the current state-of-the-art R3 adaptive sampling method, PaRS-PINN achieves significant improvements in convergence speed, prediction accuracy, and parameter identification accuracy.

**Strengths:**

1. PaRS is the first to organically integrate regional decomposition, Pareto front analysis, dynamic weighting, and resampling into a unified framework. Particularly innovative is its use of the statistical characteristics of the Pareto front (variability + improvement rate) to construct a "significance score" for guiding weight allocation. This approach surpasses simple residual-driven sampling by capturing inter-regional competition and global trade-offs, representing an important extension of the PINN training paradigm.

2. The method is well-motivated, with detailed algorithmic and formula descriptions provided for each component (decomposition, MOO, PGW, resampling). The experimental design is comprehensive: it covers three types of typical problems (parabolic PDE, hyperbolic PDE, stiff ODE), compares against strong baselines (uniform sampling and R3), and includes ablation analysis through loss/weight evolution plots. Hyperparameter settings are transparent, and results are validated through multiple repeated experiments. The appendix provides exhaustive implementation details, enhancing credibility.

**Weaknesses:**

1. Although Appendix C shows that a single PaRS update incurs minimal overhead (adding only 11 seconds for the convection equation), the NSGA-II loop (with population size Np=50 and generations Ng=100) requires 5,000 forward passes per round. For high-dimensional PDEs or large networks, this overhead may become prohibitive. While parallelization is mentioned, scalability beyond 2D problems is not demonstrated.

2. Using K-means to cluster collocation point coordinates implicitly assumes that "spatial proximity ≈ similarity in error behavior." However, this may not hold for non-local dynamics (such as wave propagation) or complex geometries. No comparisons were made with other partitioning methods (such as residual-based clustering or fixed grids), nor was the sensitivity of the number of regions M (fixed at 3) analyzed.

3. The main comparison is limited to R3, while other adaptive methods (such as RAR-D and DMIS) also deserve attention. The lack of these comparisons weakens the claim of "outperforming SOTA."

**Questions:**

1. How does PaRS perform on 3D PDEs (such as Navier-Stokes) or high-dimensional ODE systems? Can more lightweight Pareto estimation methods (e.g., gradient-based approximations) replace NSGA-II to reduce computational costs?

2. Has non-spatial clustering (e.g., based on residual magnitude or gradient norms) been attempted? Has dynamic adjustment of the number of regions M (such as adaptive splitting/merging) been considered, rather than fixing it at 3?

3. Can comparative experiments with RAR-D and DMIS be supplemented? This would more comprehensively demonstrate the advancement of PaRS.

4. What are the respective contributions of variability (σj) and improvement rate (rj) in the "significance score"? How much would performance degrade if only one of them were used?

5. All experiments are inverse problems (with unknown parameters). Is PaRS equally effective in pure forward problems (without parameter estimation)? This would test its general applicability.

---

### Official Review · Reviewer_hJDD · 2025-10-28

**Soundness:** 2
**Presentation:** 3
**Contribution:** 2
**Rating:** 4
**Confidence:** 4

**Summary:**

This paper presents a Pareto-guided regional sampling (PaRS) framework to tackle the imbalance optimization problem during the training of PINN. Specifically, PaRS treats multiple subregions of the spatial domain as competing objectives, where the subregion splitting is constructed based on K-means clustering and the regions are subsequently balanced based on Evolutionary Loop. After that, the Pareto-guided weighting and weighting-informed sampling are adopted to help the optimization. As a result, PaRS demonstrates a better performance than the R3 method.

**Strengths:**

-	It is interesting to treat the PINN optimization problem as a multiple-region competing problem. And the proposed method seems reasonable.

-	This paper is well written, and the method part is clear.

-	Model performance is better than R3.

**Weaknesses:**

### (1) The design seems combinational.

Although the motivation in balancing multiple regions is reasonable, the concrete design is a combination, which is a combination of K-means clustering, NSGA-II algorithm, and weight sampling technique. It is hard to identify the novelty of this paper’s technical design.

### (2) Missing baselines.

Since the authors focus on the region optimization and loss reweighting, I think the following baselines should be included.

[1] Respecting causality for training physics-informed neural networks, Computer Methods in Applied Mechanics and Engineering 2024

[2] RoPINN: Region Optimized Physics-Informed Neural Networks, NeurIPS 2024.

[3] When and why pinns fail to train: A neural tangent kernel perspective. Journal of Computational Physics, 2022

### (3) Missing ablations.

Since the PaRS is kind of combinational, it is very important to identify the contribution of each component in the final performance. I think the authors should provide some ablations, such as removing the weighting-informed sampling.

Besides, I think K-means clustering is time-consuming; the efficiency comparison (running time and GPU memory) w.r.t. R3 and baseline is expected.

### (4) Missing visualization of the subregion splitting and adaptive sampling.

If the authors can provide the visualization of subregion splitting and adaptive sampling results during the training phase, it will make the method more interpretable.

**Questions:**

Please see the weakness.

---

### Note · Authors · 2025-12-19

I have read and agree with the venue's withdrawal policy on behalf of myself and my co-authors.